# Predictive Relationship between Sustainable Organisational Practices and Organisational Effectiveness: The Mediating role of Organisational Identification and Organisation-Based Self-Esteem

**Chiyem Lucky Nwanzu** [1] **and Sunday Samson Babalola** [2,*] 

[1] Department of Psychology, Delta State University, Abraka 330106, Nigeria; nwanzuchiyem@gmail.com
[2] School of Management Sciences, University of Venda, Thohoyandou 0950, South Africa
[*] Correspondence: sunday.babalola@univen.ac.za

**Abstract:** This study ascertains the predictive relationship that sustainable organisational practices have with organisational effectiveness and the mediating role of organisational identification and organisation-based self-esteem in this relationship. One-hundred and forty-five participants (62 males and 83 females) were sampled from 31 privately-owned organisations in Delta State, Nigeria. Regression analysis revealed that sustainable organisational practices positively and significantly predict organisational effectiveness, $\beta = 0.42$, $p < 0.001$, and that organisational identification and organisation-based self-esteem mediate the relationship. It was recommended that privately-owned organisations intensively implement sustainable organisational practices for organisational effectiveness, organisational identification and organisation-based self-esteem.

**Keywords:** sustainable practices; organisational effectiveness; organisational identification; organisation-based self-esteem; mediating

## 1. Introduction

The necessity for organisational effectiveness is abundantly clear from theory and research. Sometimes silent, organisational effectiveness is the drive for every theory of organisation, the critical question in any form of organisational analysis [1], and the key dependent factor in organisational research [2]. Clearly, organisations that are ineffective head towards entropy, decline and death. Organisational effectiveness is strongly desired and vigorously pursued, while it is controversial and difficult to conceptualise. The issues surrounding conceptualisation arise largely because the various stakeholders (shareholders, employees, management, suppliers, and regulating agencies, among others) of organisations have differing, sometimes conflicting expectations from organisations. However, organisational effectiveness could be seen as the degree to which an organisation achieves its stated goals, acquires the resources needed, functions with minimum internal strain and meets the needs and expectations of its stakeholders. This is an eclectic definition, as it directly or indirectly implicates every extant model of organisational effectiveness.

A scholastic and lay concern for determinants of organisational effectiveness is historical. Informed by theories, scholastic research has been extended to sustainable organisational practices. Sustainable organisational practices are about organisations operating in the interest of all current and future stakeholders in a manner that ensures the long-term health and survival of the business and its associated economic, social, and environmental systems [3]. It encompasses organisational-level practices that are economically, environmentally, and socially responsible. Such practices could reflect

how the building of the organisation is designed, employees are selected and promoted, and goods and services are produced, packaged, distributed, and disposed of [4].

Organisational identification refers to an employee's perceived oneness with or belongingness to an organisation, resulting in this employee defining himself/herself in terms of the organisation of which he or she is a member [5]. It is characterised by the perception of being the same as other members of the organisation and a feeling of commonality with the organisation and support from the organisation [6]. Social identity theory provides the theoretical explanation for organisational identification. According to this theory, people create their self-concepts through their affiliation, relation, and connection with a specific social group, while identification with such a social group is determined by their behaviour in terms of their group membership [7]. Organisational identification could be expressed cognitively and emotionally, such as by internalising organisational values and being proud to be a member of an organisation.

Pierce, Gardner, Cumming, and Dunham [8] conceptualised organisation-based self-esteem (OBSE) as the degree to which an employee believes he/she is important, meaningful, effectual, and worthwhile in the organisation. Self-consistency theory [9], among others, offer an explanation for OBSE. As proposed in the theory, people strive to maintain a positive self-perception. Consequently, employees with a high OBSE will have behaviours that their organisations value in relation to maintaining their self-concept. OBSE is influenced by factors, such as organisational structure, employee participation programs, management credibility, organisational and co-workers support, enriched work, and adequate resources [10].

Ecological modernization, natural resource-based and social exchange theories propose that sustainable organisational practices have the potential to positively impact organisational effectiveness. However, it is evident in the literature that there is a dearth of studies that examine the influence of sustainable organisational practices on organisational effectiveness. This is not unexpected, as the concept of sustainable organisational practices is relatively recent in organisational behaviour literature. Added to the dearth of study on the relationship is that none of the very few existing studies examined the mediating role of work attitudes. The absence of such studies is a critical gap in the literature, as work attitudes have been widely reported to mediate and moderate the relationship between some organisational variables. To fill this gap, this study ascertained whether sustainable organisational practices predicts organisational effectiveness and whether organisational identification and OBSE mediate the relationship.

As proposed in a few theories, sustainable organisational practices have the potential to influence not only sustainable development, but also a number of desirable organisational variables, including organisational effectiveness. This implies that organisations need some degree of the practices. This study therefore aims to examine how many sustainable organisational practices exist in privately-owned organisations. Organisational effectiveness is a highly sought organisational outcome. This largely explains the numerous theoretical and empirical efforts to identify its determinants. However, while a number of variables that theoretically seem to positively influence organisational effectiveness have attracted extensive research, a few others, including sustainable organisational practices, have very scanty studies on them. In addition, this study aims to add to the number of studies that provide knowledge on how sustainable organisational practices relate to organisational effectiveness. Studies on the influence sustainable organisational practices on organisational effectiveness are not only few, but they failed to examine the moderating and mediating roles of work attitudes. As it is with some other determinants of organisational effectiveness, a few work attitudes could mediate the influence of sustainable organisational practices on organisational effectiveness. Consequently, this study provides knowledge on whether organisation-based self-esteem and organisational identification have mediating roles in the relationship. The specific objectives accumulate in the overall objective of this study, which is to provide an understanding that would be of practical value to privately-owned organisations in utilizing sustainable organisational practices, organisational identification, and OBSE for organisational effectiveness.

The ecological modernisation theory draws attention to ecological criteria in the design, performance and evaluation of production processes [11], such as that of Chen [12], who opines that sustainability practices can enhance innovation opportunities in terms of organisation product and process innovation, which can lead to the achievement of economic profitability. The natural resource-based theory emphasised competitive advantage, based upon the firm's relationship to the natural environment, such as product stewardship and sustainable development [13]. These theories assert that organisations that promote and sustain good relationships with the ecosystem would achieve a sustainable competitive advantage from the efficient use of natural resources [12]. Social exchange theory involves a series of interactions that generate obligations [14], with these interactions seen as interdependent and dependent on the action of other entities. Central to the theory is the norm of reciprocity that obligates individuals to respond positively to favourable treatment received from another entity [15]. By implication, when employers provide their employees with positive work experiences, the employees will experience organisational identification and OBSE.

Ecological modernisation theory and natural resource-based theory explain the hypothesised influence of sustainable organisational practices on organisational effectiveness, while social exchange theory explains two sets of relationships. First, it explains the relationship between sustainable organisational practices, organisational identification, and OBSE. Second, it explains the relationship between organisational identification and OBSE organisational effectiveness. In other words, sustainable organisational practices benefit employees, either as members of an organisation or members of society. Employees could respond to the benefits by expressing organisational identification and OBSE, and these work attitudes are well documented in predicting other organisational variables that have organisational effectiveness as the outcome.

Empirical study on sustainable organisational practices, and its relationship with organisational effectiveness and the mediating and moderating variables in the relationship, is in infancy. However, the very few extant studies point at the desirable influence of sustainable organisational practices on organisational performance. For instance, Maletic, Maletic, Dahlgaard, Dahlgaard-Park, and Gomiscek [16] studied the relationship between sustainable organisational practices and organisational performance, and the mediating role of non-financial performance outputs in the relationship with data collected from organisations from Germany, Poland, Serbia, Slovenia, and Spain and reported that innovation performance exerts a mediation effect in the relationship. Siew, Balatbat, and Carmichael [17] examined the relationship between sustainability practices and the financial performance of 44 construction companies and observed that the group that have non-financial reports outperform other groups. The study used secondary data [18] listings, and because the data were the companies' presentation of themselves to the public, there could be issues of positive self-presentation. Again, as with similar studies, the measure of organisational performance was limited to financial indicators (profitability and equity valuation).

Chen's [12] investigation of the sustainable initiatives of manufacturing companies, relationship between sustainable practices and the companies' performances in finance, operation, innovation, environment, and society showed that a positive relationship existed. Gomišček and Maletič [19] studied the total-quality management's sustainability-oriented innovation practices and their contribution to organisational performance among 166 managers from Slovenian organisations, indicating positive relationships, although the organisational performance measure lacks the necessary rigour and processes of scale construction. None of the studies cited above examined the individual variable, as a mediator in the relationship between sustainable organisational practices and organisational performance. The less frequent use of individual variables on how sustainable organisational practices influence organisational performance is a weakness, as there is no understanding of the individual-level variables in the process.

OBSE is widely observed to influence a number of other organisational variables in desirable directions. For instance, OBSE significantly influences employee work engagement [20], organisational commitment [21], employee performance [22], the spirit at work [23], innovative work behaviour [24],

and job satisfaction [25]. However, it significantly moderates the association among incentive motivators and employee performance [22]. Organisational identification is also observed as a valuable organisational variable. For instance, the centrality and continuity dimensions of organisation identification positively influence organisational objectives [26]. Oktug's [27] study showed a positive relationship between organisation identification and job satisfaction, while Qureshi, Shahjehan, Zeb, and Saifullah [28] reported that self-esteem and organisational identification are significant predictors of organisational citizenship behaviour. The preceding theoretical and empirical review the research framework is shown in Figure 1.

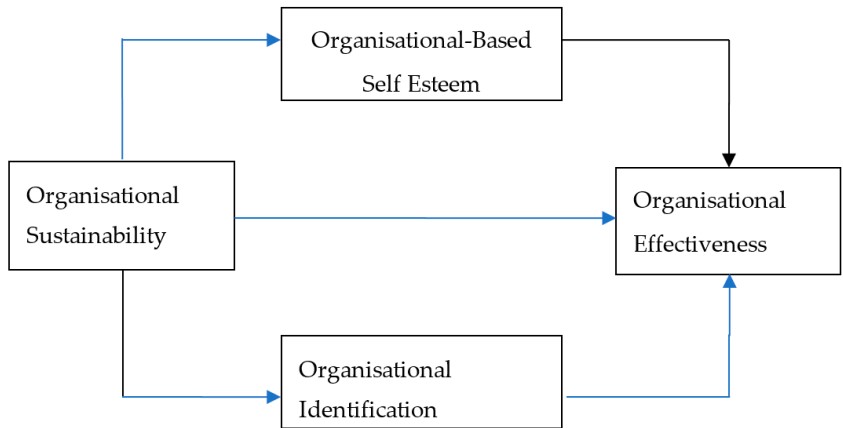

**Figure 1.** Conceptual framework, depicting the relationship between the studied variables.

*Hypotheses*

To test the identified model, derived from the theoretical and empirical reviews, four hypotheses are generated, which are as follows:

**Hypotheses 1 (H1):** *Sustainable organisational practice have a positive and significant predictive relationship with organisational effectiveness.*

**Hypotheses 2 (H2):** *Organisational identification mediates the predictive relationship between sustainable organisational practice and organisational effectiveness.*

**Hypotheses 3 (H3):** *OBSE mediates the predictive relationship between sustainable organisational practice and organisational effectiveness.*

**Hypotheses 4 (H4):** *Organisational identification and OBSE mediate the predictive relationship between sustainable organisational practice and organisational effectiveness.*

## 2. Materials and Methods

### 2.1. Participants

One-hundred and forty-five participants, sampled from 31 privately-owned organisations in Delta State, Nigeria, provided the analysed data. Like every other society, sustainable development is of concern to Delta State. Therefore, it is important for the State to have an understanding of its potential for sustainable development. This knowledge would be revealed in studies on the sustainable behaviour of individuals and organisations, as sustainable development is a function of the behaviour of the two entities. The sampled organisations include 9 banks, 16 educational institutions (secondary schools and one university), 4 hotels, and 2 Global System for Mobile Communications (GSM) providers. For comparative studies, the use of a large number of samples organisations and a few respondents in each organisation yields results with a greater degree of external validity than otherwise [29]. The adopted sample size has 80% power at $p < 0.05$, when the effect size is medium [30].

The respondent sample comprises 62 males and 83 females, and 96% of the participants hold a bachelor or post-graduate certificate. Their age $\bar{x}$ = 33.66 (SD = 6.78, Range, 26). The sampled organisations include small-size, 25%, medium-size, 42%, and large-size, 32%, organisations. This statistic was based on the common criteria of 10 to 49, 50 to 249, and 250 and above workforces, as small, medium and large-size organisations, respectively [31]. Thirty, 42, and 73 participants were drawn from the small-size, medium-size, and large-size organisations, respectively. The participants were drawn from both managerial and non-managerial staff members, as a combination of perspectives and understanding from different internal stakeholders better represent the prevailing situation in the organisations.

*2.2. Instruments*

Sustainable organisational practices were measured with items adapted from Harmon, Fairfield, and Behson's [32] sustainable organisational practices scale and Cella-De-Oliveira's [33] organisational sustainability indicators. Adapting items from the two sources maximizes the strength and minimizes the weakness of each measure. For instance, the scale of Harmon et al. [32] lacks item on economic aspect of sustainable organisational practices, and some of the items were too inclusive. Cella-De-Oliveira's [33] indicators were presented under the three dimensions of sustainable organisational practices. This guided the present researcher in grouping the scale items under environmental and social dimensions. Sample items on the scale are "In the organisation where I work, there are practices that ensure the reduction in waste materials", and "In the organisation where I work there are practices that ensure salary equality between genders within the limits of each post". Nwanzu and Uhiara's [34] 40-item measure on organisational effectiveness was adopted. The scale was developed in four models (goals, systems resources, internal processes, and stakeholders) of organisational effectiveness. Sample items from the scale are "In the organisation where I work, the desired input-output ratio is attained all the time", "In the organisation where I work, the interests of the various constituencies are often satisfied", and "In the organisation where I work, there is job satisfaction among employees".

Organisational identification was measured with Mael and Ashforth's [35] six-item scale. A sample item from the scale is "when someone criticises the organisation I work for, it feels like personal insult". Organisation-based self-esteem was measured with Pierce et al.'s [8] 10-item scale. A sample item from the scale is "I am taken seriously in the organisation I work for". A five-point Likert method of summated the rating scale—strongly agree (5), agree (4), undecided (3), disagree (2), strongly disagree (1)—was adopted, as it generates enough variability in the response. Generating sufficient variance among respondents through scaling gives validity to the statistical outputs [36]. Wide scale points also control the effects of the central tendency, i.e., the tendency of respondents to avoid the extreme end of scales. For all the scales, scores we computed by averaging each participant's response to the items. Cronbach's alpha reliability coefficient, $\alpha$ = 0.51, for sustainable organisational practices, $\alpha$ = 0.83, for organisational effectiveness, $\alpha$ = 0.68, for organisational identification, and $\alpha$ = 0.93, for OBSE, were obtained for this study. These statistics indicate that the scales, except for sustainable organisational practices, are of good reliability; $\alpha$ = 0.70 or above is considered satisfactory [37].

*2.3. Procedure*

Data were collected in participating organisations through the convenience sampling technique, the most common form of non-probability sampling technique [38]. It is a convenience sample, because organisations and respondents that met the criteria for participation were used on the basis of availability. For instance, every participant in each sampled organisation had served for a period of not less than two years. It was assumed that a period not less than two years in an organisation is long enough for employees to understand the prevailing situation in their organisations. The above timeframe aligned with the prescription of Martz [39], and it is the most practical way of assessing organisational effectiveness to consider a time frame of one to five years; anything less than one year may not fully reflect the contribution of various strategies and initiatives that require some period of maturation to show an effect. The convenience sampling technique was adopted because of the issue

of the sampling frame. As Bryman [40] noted, random sampling is unlikely to be feasibly when there is no sampling frame or when the frame would be absurdly expensive or even impossible to construct.

*2.4. Design*

The design was cross-sectional, as the sample was drawn from the population, and data were collected from the sample at one point in time [38]. A few reasons informed the choice of this research design. First, the level of analysis was the organisation-level, and to assess enough organisations that would form a statistically satisfactory sample, a questionnaire-based field survey design becomes very appropriate. Second, hypotheses of this study were in generalized and sweeping forms (e.g., sustainable organisational practices will positively and significantly predict organisational effectiveness). This structure of presentation provides results that have a wide coverage. Therefore, the potential for result generalisation that is associated with the survey makes it very suitable for this study. As expressed by Holton and Burnelt [41], surveys enable one to use smaller groups of people to make inferences about larger groups that would be prohibitively expensive to study. Data were collected and analysed at the organisational level. This was achieved through the wording of the questionnaire items. Every item on the independent, mediating and dependent variables started with the phrase "In the organisation where I work". The choice of organisational level analysis was basically informed by the widely reported inadequacy of aggregated responses as a measure of organisational properties [42]. Just as a group is more than or different from the sum of their individual members (synergy), so is an organisation more than or different from the sum of the individual employees.

*2.5. Data Analysis*

Hypothesis 1 was tested with regression analysis, while Hypotheses 2, 3, and 4 were tested with partial correlation (as recommended in Howitt and Crammer [37]). Partial correlation is the correlation between a pair of variables, after adjusting for the effect of a third variable [43]. The adopted statistical tests are parametric; therefore, assumptions associated with their usage were taken into account. For instance, the collected data were independent of each other. The Likert scaling format was used to achieve interval scaling. Scatter plots, produced with IBM-SPSS from the data, showed a linear relationship between each pair of variables. IBM-SPSS Version 25 was used for data analysis.

## 3. Results

Descriptive statistics revealed a high degree of sustainable organisational practices, organisational effectiveness, organisational identification, and OBSE. With a five-point Likert summated rating scale, $\bar{x} = 4.10$ ($SD = 0.49$), $\bar{x} = 4.07$ ($SD = 0.53$), $\bar{x} = 4.26$ ($SD = 0.55$), and $\bar{x} = 4.32$ ($SD = 0.54$) were observed for the variables. The statistics, as shown in Table 1, revealed a positive and significant correlation between the variables. The degree of correlation between the predictor, the moderator and the criterion variables were modest, indicating the absence of multicollinearity in the model.

**Table 1.** Correlation matrix for OSP, OE, OBSE, and OI.

|        | OSP     | OE      | OBSE    | OI  |
|--------|---------|---------|---------|-----|
| OSP    | 1       |         |         |     |
| OE     | 0.42 ** | 1       |         |     |
| OBSE   | 0.25 ** | 0.46 ** | 1       |     |
| OI     | 0.48 ** | 0.53 ** | 0.53 ** | 1   |

Note: OSP = sustainable organisational practices; OE = organisational Effectiveness; OBSE = organisation-based self-esteem; OI = organisational identification. ** $p < 0.01$.

Statistics at the base of Table 2 shows a simple regression analysis, predicting organisational performance from sustainable organisational practices. The *R* value (0.42) indicates that, as scores on sustainable organisational practices increase, the score on organisational effectiveness increases.

This is a significant positive correlation, β = 0.42, *p* < 0.001. The $R^2$ indicates that sustainable organisational practices account for 18 percent of the variance in organisational effectiveness. On the basis of Cohen's [44] criterion, $R^2$ of 0.18 indicates a medium effect size. The analysis of variance (ANOVA) test, $F$ (1, 145) = 31.72, *p* < 0.001, also indicated that the regression is statistically significant. Table 2 shows a partial correlation for the mediating role of organisational identification and OBSE in the relationship between sustainable organisational practices and organisational effectiveness. The correlation between sustainable organisational practices and organisational effectiveness was r = 0.42, *p* < 0.001. The first-order correlation, when controlling for OBSE, declined to r = 0.36, which is also significant, *p* < 0.001. The smaller first-order correlation, when compared to the zero-order correlation, indicates that OBSE has a mediating effect on the relationship.

**Table 2.** Partial correlation on mediating role of OI and OBSE in the relationship between OSP and OE.

| Predictor Variable | Criterion Variable | Zero-Order Correlation | Controlled Variable | First and Second Order Correlation |
|---|---|---|---|---|
| OSP | OE | 0.42 ** | OBSE | 0.36 ** |
| OSP | OE | 0.42 ** | OI | 0.22 * |
| OSP | OE | 0.42 ** | OBSE & OI | 0.23 * |

Note: OSP = sustainable organisational practices; OE = organisational effectiveness; OBSE = organisation based self-esteem; OI = organisational identification. ** = *p* < 001; * = *p* = 0.01; $R$ = 0.42; $R^2$ = 0.18; Adjusted $R^2$ = 0.17, n = 145, $F$ = 31.72; *p* < 0.001.

However, the relationship remains significant as the correlation (zero-order) between sustainable organisational practices and organisational effectiveness was r = 0.42, *p* < 0.001, even when OBSE was controlled. The first-order correlation, when controlling for organisational identification, declined to r = 0.22, which is also significant, at *p* < 0.006. The smaller first-order correlation, when compared to the zero-order correlation, indicates that organisational identification has a mediating effect on the relationship. However, the relationship remains significant, even when organisational identification was controlled. The correlation between sustainable organisational practices and organisational effectiveness was r = 0.42, *p* < 0.001. The second-order correlation, when controlling for OBSE and organisational identification, declined to r = 0.23, which is also significant, at *p* < 0.005. The smaller second-order correlation, when compared to the zero-order correlation, indicates that OBSE and organisational identification have a mediating effect on the relationship. However, the relationship remains significant, even when organisational identification and OBSE were controlled. The zero-order correlation of r = 0.42 indicates a large effect size.

The findings from the study is graphically presented in Figure 2.

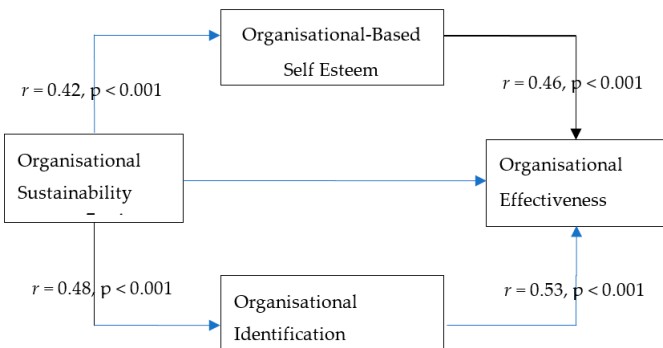

**Figure 2.** Depict empirical the relationship between the studied variables.

Multiple regression analysis (Table 3) shows organisational effectiveness predicted from environmental and social β dimensions of sustainable organisational practices. The two dimensions significantly predicted organisational effectiveness. Specifically, environmental dimension,

($\beta$ = 0.43, *p* < 0.001) and social dimension, ($\beta$ = 0.08, *p* < 0.001). Part correlation revealed that environmental dimension and social dimensions account for 40% and 7% variance in organisational effectiveness, respectively.

**Table 3.** Multiple regression analysis predicting OE from OSPE and OSPS.

| | **B** | *t* | **Part Correlation** | **P** | **95% CI** | |
| --- | --- | --- | --- | --- | --- | --- |
| | | | | | **Lower Limit** | **Upper Limit** |
| OSP (E) | 0.43 | 5.42 | 0.40 | 0.001 | 0.28 | 0.60 |
| OSP(S) | 0.08 | 1.01 | 0.07 | 0.001 | −0.05 | 0.17 |

Note: OSPE = sustainable organisational practices (environmental); OSPS = sustainable organisational practices (social).

Table 4 shows partial correlations on the mediating role of organisational identification and OBSE in the relationship between environmental and social dimensions of sustainable organisational practices and organisational effectiveness. The correlation (zero-order) between environmentally sustainable organisational practices and organisational effectiveness was r = 0.46, *p* < 0.001. The first-order correlation when controlling for OBSE decline to 0.39, which was also significant at *p* < 0.001. The smaller first-order correlation when compared to the zero-order correlation indicates that OBSE has a mediating effect on the relationship. However, the relationship remains significant even when OBSE was controlled.

**Table 4.** Partial correlation for the mediating role of OI and OBSE in the relationship between the dimensions of OSP and OE.

| **P Variable** | **C Variable** | **Zero-Order Correlation** | **Controlled Variable** | **First and Second Order Correlation** |
| --- | --- | --- | --- | --- |
| OSP(E) | OE | 0.46 ** | OBSE | 0.39 ** |
| OSP(E) | OE | 0.46 ** | OI | 0.35 ** |
| OSP(E) | OE | 46 ** | OI & OBSE | 0.33 ** |
| OSP(S) | OE | 0.25 * | OBSE | 0.21 * |
| OSP(S) | OE | 0.25 * | OI | 0.01 |
| OSP(S) | OE | 0.25 * | OI & OBSE | 0.05 |

Note: OSPE = sustainable organisational practices (environment); OSPS = sustainable organisational practices (social); OE = organisational effectiveness. ** = *p* < 0.01; * = *p* < 0.1.

The correlation (zero-order) between environmentally sustainable organisational practices and organisational effectiveness was r = 0.46, *p* < 0.001. The first-order correlation when controlling for organisational identification declined to r = 0.35, which was significant at *p* < 0.001. The smaller first-order correlation when compared the zero-order correlation indicates that organisational identification has a mediating effect on the relationship. However, the relationship remains significant even when OBSE was controlled. Correlation (zero-order) between environmentally sustainable organisational practices and organisational effectiveness was r = 0.46, *p* < 0.001. The second-order correlation when controlling for organisational identification and OBSE declined to r = 0.33, which was also significant *p* < 0.001. The smaller second-order correlation when compared to the zero-order correlation indicates that OBSE and organisational identification have a mediating effect on the relationship. However, the relationship remains significant even when organisational identification and OBSE were controlled. Correlation (zero-order) between social sustainable organisational practices and organisational effectiveness was r = 0.25, *p* < 0.01. The first-order correlation when controlling for OBSE decrease to 0.21, which was significant at *p* < 0.01. The smaller first-order correlation when compared to the zero-order correlation indicates that OBSE has a mediating effect on the relationship. However, the relationship remains significant even when OBSE was controlled.

Correlation (zero-order) between social sustainable organisational practices and organisational effectiveness was r = 0.25, $p < 0.01$. The first-order correlation when controlling for organisational identification declined to r = 0.01, which was not significant at two tailed $p > 0.05$. Compared to the zero-order correlation, the first-order correlation was smaller and non-significant. This indicates that organisational identification has a large mediating effect on the. Finally, the correlation (zero-order) between social sustainable organisational practices and organisational effectiveness was r = 0.25, $p < 0.002$. The second-order correlation when controlling for organisational identification and OBSE declined to r = 0.05, which was not significant at two-tailed $p > 0.05$. Compared to the zero-order correlation, the second-order correlation was smaller and non-significant. This indicates that organisational identification and OBSE have a large mediating effect on the relationship.

## 4. Discussion

This study examined the predictive relationship that sustainable organisational practices have with organisational effectiveness and the mediating roles of organisational identification and OBSE in the relationship. The hypothesis that sustainable organisational practices have a significant positive predictive relationship with organisational effectiveness was supported. The result was expected, as it is consistent with the extant literature. For instance, Chen [12] investigated the degree of sustainable initiatives of manufacturing companies, and the relationship between these sustainable practices and the companies' performance and observed a positive relationship. Gomišček and Maletič [19] studied TQM sustainability-oriented innovation practices and their contribution to organisational performance and observed positive relationships between sustainability-oriented innovation practices and all organisational performance dimensions. The hypothesis that organisational identification, OBSE, and a combination of organisational identification and OBSE would positively and significantly predict organisational effectiveness was partly supported. It was judged to be partly supported, as the relationships were positive but non-significant. The zero-order correlation substantially decreased when organisational identification and OBSE, and organisational identification and OBSE combined, were controlled for. This indicates that the controlled variables are mediators in the relationship. However, that the first-order and second-order correlations remain significant, after controlling for the variables. Further analyses revealed that both the environmental and social dimensions of sustainable organisational practices have a positive and significant predictive relationship with organisational effectiveness. However, there was no significant relationship between social sustainable practices and organisational effectiveness, when controlling with organisational identification and OBSE combined. This implies that the variables of organisational identification and a combination of organisational identification and OBSE are principal mediators in the relationship.

### 4.1. Conclusions

A few conclusions can be drawn from the findings. First, on the basis of the very high mean scores obtained, it could be concluded that OI, OBSE, and sustainable organisational practices are highly expressed in the privately-owned organisations. Second, a medium effect size was observed in the relationship between organizational sustainable practices and organizational effectiveness. It could be concluded that sustainable organisational practices have a significant and important contribution to the effectiveness of organizations. Third, OI and OBSE are among the variables that mediate the relationship between sustainable organisational practices and organisational effectiveness. Forth, the first-order correlation for organizational identification was very low, when compared to the zero-order correlation. It could be concluded that organisational identification is a principal mediating variable in the relationship between sustainable organisational practices and organisational effectiveness. Fifth, compared to OBSE, OI recorded a lower first-order correlation, which indicates a superior mediating role. On that premise, it could be concluded that organisational identification is superior to OBSE, as mediators in the relationship between sustainable organisational practices and organisational effectiveness. Finally, compared to sustainable

organisational practices (social), sustainable organisational practices (environmental) have the largest influence on organisational effectiveness.

## 4.2. Recommendations for Practice

The findings of this study, which are consistent with the related extant literature, have some practical utility. Sustainable organisational practices were observed to have a desirable influence on OI, OBSE, and organisational effectiveness in privately-owned service organisations. Therefore, organisational practitioners in this type of organisation should take seriously and engage in sustainable organisational practices, not only for both the direct and indirect positive predictive relationship that sustainable organisational practices has with organisational effectiveness, but also for the direct positive relationship it has with OI and OBSE. The latter point is also necessary, as the two work attitudes (that is, OI and OBSE) are empirically well implicated in a number of desirable organisational outcomes, such as job satisfaction and work engagement [20,45].

## 4.3. Limitations and Recommendations for Future Studies

Factors that could limit the interpretation and utility of the findings are as follows. First, the design is correlational. The implication of this is that a cause–effect relationship cannot be discovered from the analysis of such data. Causality is a property occasioned by experimental design (randomization and manipulation) and to a lesser extent longitudinal study. Therefore, it is recommended that related future studies should explore field experimentation and longitudinal designs to enable causal interpretation. Second, the above data were collected using a self-report measure. Self-report measures have the potential for a social desirability bias (SDB), halo effect, and same-source variance. Therefore, future studies should adopt triangulation in data collection. Third, because the participants were drawn from a large number of organisations, the random sampling method was impracticable. Consequently, the convenience sampling method was adopted for data collection. The use of the convenience sample was a limitation to the interpretation of the results. This is because the degree to which such samples represent the population is very difficult, if not impossible, to establish. Finally, organisations are commonly grouped as either publicly-owned or privately-owned and manufacturing or service. In this study, only privately-owned service organisations were sampled. This implies that the findings cannot be validly generalised to other classes of organisations that were not sampled. However, despite these limitations, the present study is still valuable for both organisations and societies at large.

**Author Contributions:** Conceptualization, C.L.N.; Data curation, C.L.N.; Formal analysis, S.S.B.; Investigation, C.L.N.; Methodology, S.S.B.; Supervision, S.S.B.; Writing—original draft, C.L.N.; Writing—review & editing, S.S.B.

**Funding:** This research received no external funding.

**Acknowledgments:** The authors acknowledge the contribution from our universities for administrative and technical support.

**Conflicts of Interest:** The authors declare no conflict of interest.

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
