# Peer review of "Predictive Relationship between Sustainable Organisational Practices and Organisational Effectiveness: The Mediating role of Organisational Identification and Organisation-Based Self-Esteem"

_sustainability, doi:10.3390/su11123440_

Round 1
Reviewer 1 Report
The paper deals with an interesting topic and provides a good basis for interesting future research. The paper is original, written in good English, with appropriate methodology and theoretical background. The aim of the paper was clearly formulated. The structure of the paper is logical, the text is easy to read. The findings are presented and discussed. This study examined the predictive relationship organisational sustainable practices has with organisational effectiveness and the mediating roles of organisational identification and OBSE on the relationship. The hypothesis that organisational sustainable practices have a significant positive predictive relationship with organisational effectiveness was supported. The result was expected as it is consistent with the extant literature. It should be taken into account that the research still has some limitations (well described by authors) but there are a lot of potential directions for further research, so good luck! The results of further research could be a starting point for another publication (having more entities in the research sample, using additional research tools, developed comparative study).
Author Response
The paper deals with an interesting topic and provides a good basis for interesting future research. The paper is original, written in good English, with appropriate methodology and theoretical background. The aim of the paper was clearly formulated. The structure of the paper is logical, the text is easy to read. The findings are presented and discussed. This study examined the predictive relationship organisational sustainable practices has with organisational effectiveness and the mediating roles of organisational identification and OBSE on the relationship. The hypothesis that organisational sustainable practices have a significant positive predictive relationship with organisational effectiveness was supported. The result was expected as it is consistent with the extant literature. It should be taken into account that the research still has some limitations (well described by authors) but there are a lot of potential directions for further research, so good luck! The results of further research could be a starting point for another publication (having more entities in the research sample, using additional research tools, developed comparative study).
Response: Thank you for your comments, we do appreciate your suggestion to explore further studies in this area of research.
Reviewer 2 Report
Firstly, the paper objectives are not clearly defined. In this regards, is not clear why such a study would be representative? I don't understand the usefulness of the analysis, in both sides: academia and business. There are no adequate explanations and motivation regarding to the potential readers and practitioners interest in such a theme. In Introduction, which include the literature review there is an amalgam of notions presented somewhat chaotic, without a logical sequence. Expressions in the paper, in some parts (here and there), are auspicious, hard to be understand. In the paper there is some inconsistency of the terms used and even the incoherence in the logical thread of the research description.
Secondly, it is not very clear how the companies were selected and to what extent the ones presented in the study are prepresentative at national level, geographically, covers various fields and so on. Regarding the respondents, I understood that they were integrated in the study basen on availability, but it is not clear how you found this availability (is not clear the mechanism!). It does not seem very clear about the total number of respondents from where they came from: small, medium-sized or large firms (for get a distribution of these respondents related to the total respondents number:145) and at what level those respondents came from (decision-making/managerial level or operational area). Also, there is no mention of the design of the questionnaire without any kind of explanation (who design the questionnaire, if there are covered both sides: managerial and psychologycal point of view, types of questions, topics included, number of questions, time for responding etc.).
Thirdly, related to the abbreviations used at page two appeared for the first time: OBSE, but the meaning was inserted just in page 7.
Also, between the line 155 and 156 is inserted o very poorly constructed figure, without numbering and naming, without a direct reference in the paper text.
Calculated or determined elements can be considered now as clasical aspects of this kind of analysis: nothing spectacular, nothing innovative or special, as a personal touch.
Another questions is related to the contents for the tables 3.3 and 3.4. When we speak about variables: OSPE is not EOSP? and OSPS is not SOSP?
There is not a distinctive paragraphs dedicated to Conclusion. These findings are included in the Discussion are? Why? The conclusion are questionable?
On the other hand, in the Discussion paragraphs is declared that the analysis results were expected to be obtained in accordance with existing literature!! Why? Is mandatory? This idea creates more credibility for conclusions, for the study itself, for the authors... ?! If the results infringed literature then what would have been the outcome of the conclusions?
A big disadvantage, which may even alter the research results, is the non-use of the random sampling method. In this sense, it was necessary to express more clearly and resounding the limitations of the study itself.
Author Response
Firstly, the paper objectives are not clearly defined. In this regards, is not clear why such a study would be representative? I don't understand the usefulness of the analysis, in both sides: academia and business. There are no adequate explanations and motivation regarding to the potential readers and practitioners interest in such a theme. In Introduction, which include the literature review there is an amalgam of notions presented somewhat chaotic, without a logical sequence. Expressions in the paper, in some parts (here and there), are auspicious, hard to be understand. In the paper there is some inconsistency of the terms used and even the incoherence in the logical thread of the research description.
Secondly, it is not very clear how the companies were selected and to what extent the ones presented in the study are representative at national level, geographically, covers various fields and so on. Regarding the respondents, I understood that they were integrated in the study base on availability, but it is not clear how you found this availability (is not clear the mechanism!). It does not seem very clear about the total number of respondents from where they came from: small, medium-sized or large firms (for get a distribution of these respondents related to the total respondents number:145) and at what level those respondents came from (decision-making/managerial level or operational area). Also, there is no mention of the design of the questionnaire without any kind of explanation (who design the questionnaire, if there are covered both sides: managerial and psychological point of view, types of questions, topics included, number of questions, time for responding etc.).
Thirdly, related to the abbreviations used at page two appeared for the first time: OBSE, but the meaning was inserted just in page 7.
Also, between the line 155 and 156 is inserted o very poorly constructed figure, without numbering and naming, without a direct reference in the paper text.
Calculated or determined elements can be considered now as classical aspects of this kind of analysis: nothing spectacular, nothing innovative or special, as a personal touch.
Another questions is related to the contents for the tables 3.3 and 3.4. When we speak about variables: OSPE is not EOSP?andOSPS is not SOSP?
There is not a distinctive paragraphs dedicated toConclusion. These findings are included in the Discussion are? Why? The conclusion are questionable?
On the other hand, in theDiscussionparagraphs is declared that the analysis results were expected to be obtained in accordance with existing literature!! Why? Is mandatory? This idea creates more credibility for conclusions, for the study itself, for the authors... ?! If the results infringed literature then what would have been the outcome of the conclusions?
A big disadvantage, which may even alter the research results, is the non-use of the random sampling method. In this sense, it was necessary to express more clearly and resounding the limitations of the study itself.
Response: We appreciate comments of this reviewer as we view the issues raised as very germane and good towards improving and enriching the quality and clarity for easier comprehension of the readers. We have however address the issues of concerns:
1 In line 12 the word “of” between “relationship and organisational” was deleted
2 In line 15 “Delta State, Nigeria” was added.
3 The figure between line 154 and 165 was reconstructed
4 In line 166 “title” was given to the figure
5 Sub-heading “Participants” was revised
6 Sub-heading “Instruments” was revised
7 Sub-heading “Procedure” was revised
8 Abbreviation of OBSE in p. 2, was first presented full in that p. 2 before the abbreviation. See line 55.
9 Lines 180-187 were revised to include information on the number of the different types of organizations sampled
10 The percentage of small, medium and large size organizations in the total population was presented in line 192 to 198
11 Sample item(s) from each scale are now added to the manuscript
12 In line 206 “research” was changed to “scale”
13 The authors of the five scales adapted in the study were mentioned and the number of items for each scale was also mentioned in line 207 to 218
14 In line 274 “Table 1” was changed to “Table 3.1”
15 In line 272 this statement was added “The degree of correlation between the predictor, the moderator and the criterion variables were modest; indicating the absence of multicollinearity in the mode”.
16 Table 3.1 that was wrongly positioned was appropriate positioned
17 The presentation of the variables in Tables 3,3 and 3,4 have been corrected from “Environmental Organizational Sustainable Practices - EOSP” To Organizational Sustainable Practices (Environment) OSPE, “Social Organizational Sustainable Practices - SOSP” to Organizational Sustainable Practices (Social) OSPS
18 Table titles were revised
19 Figures 2 added
20 Three subsections - “Conclusion”,” Recommendation for Practice” and Limitation and Recommendation for Future Studies” were created from the ‘Discussion Section” therefore:
21 Four citations and references added
Reviewer 3 Report
Dear Authors,
The article is interesting and well-founded. It raises important issues. A well done literature review should be appreciated. The results of the research are presented in an interesting way. Triangulation would be justified in the further research process.
Pay attention to the figure. There is no reference to the figure. The drawing should be graphically corrected and signed. An introduction to the hypotheses can also be made.
I hope that small remarks will be useful.
Author Response
The article is interesting and well-founded. It raises important issues. A well done literature review should be appreciated. The results of the research are presented in an interesting way. Triangulation would be justified in the further research process.
Pay attention to the figure. There is no reference to the figure. The drawing should be graphically corrected and signed. An introduction to the hypotheses can also be made.
I hope that small remarks will be useful.
Response: The authors wish to appreciate the reviewer and the need for further research processes from triangulation point of view.
In lines 152 -169, The figure has been clearly checked and corrected, reference to the figure has now been provided, hypotheses introduced, with the drawing graphically corrected
Round 2
Reviewer 2 Report
I appreciate that you did your best to improve the quality paper and respond to major comments. Still there are some things that you need to change/modify.
You have two figures. For the first one, you need to decide what kind of format numbering you will use. Since the instructions for the authors mention only one number I consider that you have to use also for the tables this way of numbering. You can avoid a lot problems. In the line 153 you mentioned figure 1 but after the figure you inserted figure 1.1 (line 166). Another problem is that for the second figure you use figure 3.1 (line 313)?!. Is better to use figure 2. After the figure you don't have the figure number and content declared. These figures need to be centerd on the middle.For the second figure the r from the down left corner has no comma(point) before the numbers and the p is wrote with capital letter. Also the p's from the right part from this figure don't have the comma (point) before the numbers. For both figures I think it should be use the same color in drawing the lines (not blue and black). Also is better to use only one number for the tables (Table 1, Table 2 ..).
On the rows 192 and 193 there are used different size for the fonts! Please correct it accordingly!
Also, check the rows 293 and 294 (there is an interruption of continuity)!
In the subchapter 2.5, line 258, are mentioned hypotheses 1 to 4. Therefore, starting with the line 170 you need to number all of these four hypotheses. Otherwise it could appear some misundertanding within the paper.
Another place of misunderstanding is related to the notes for the tables 3.3 and 3.4. In the table 3.3 you have declared OSPE = Sustainable organisational practices(Environmental) while in table 3.4. the same OSPE is declared differently: Organisational sustainable practice(Environment) !?
Finally, in the references area, line 508, to the new added title you numbered twice (31 two times!!!).
In general, you need to be more careful with these kind of details!!!!!
Author Response
We do appreciate the fine detailed comments from the review. We will take note of correcting such overlooked detailed in future communications. We have however corrected the identified errors in reporting.
You have two figures. For the first one, you need to decide what kind of format numbering you will use. Since the instructions for the authors mention only one number I consider that you have to use also for the tables this way of numbering. You can avoid a lot problems. In the line 153 you mentioned figure 1 but after the figure you inserted figure 1.1 (line 166). Another problem is that for the second figure you use figure 3.1 (line 313)?!. Is better to use figure 2. After the figure you don't have the figure number and content declared. These figures need to be centred on the middle. For the second figure ther from the down left corner has no comma(point) before the numbers and thepis wrote with capital letter. Also thep's from the right part from this figure don't have the comma (point) before the numbers. For both figures I think it should be use the same colour in drawing the lines (not blue and black). Also is better to use only one number for the tables (Table 1, Table 2 .).
The correction on the figure presented in Figure 2.1 has been done. All correction in this work done are presented in red.
On the rows 192 and 193 there are used different size for the fonts! Please correct it accordingly!
Font sizes are now similar throughout the write up.
Also, check the rows 293 and 294 (there is an interruption of continuity)!
The sentence is now rephrased.
In the subchapter 2.5, line 258, are mentioned hypotheses 1 to 4. Therefore, starting with the line 170 you need to number all of these four hypotheses. Otherwise it could appear some misunderstanding within the paper.
Hypotheses are now numbered
Another place of misunderstanding is related to the notes for the tables 3.3 and 3.4. In the table 3.3 you have declared OSPE = Sustainable organisational practices(Environmental) while in table 3.4. the same OSPE is declared differently: Organisational sustainable practice(Environment) !?
Similar words are now use in both tables 3.3 and 3.4.
Finally, in the references area, line 508, to the new added title you numbered twice (31 two times!!!).
Figure 31 written twice have been edited correctly.